

# Temporal network embedding framework with causal anonymous walks representations

Ilya Makarov[1,2,3], Andrey Savchenko[4], Arseny Korovko[1], Leonid Sherstyuk[1], Nikita Severin[5], Dmitrii Kiselev[1,3], Aleksandr Mikheev[1,6] and Dmitrii Babaev[3,6]

[1] HSE University, Moscow, Russia
[2] University of Ljubljana, Ljubljana, Slovenia
[3] Artificial Intelligence Research Institute (AIRI), Moscow, Russia
[4] Laboratory of Algorithms and Technologies for Network Analysis, HSE University, Nizhny Novgorod, Russia
[5] Moscow Institute of Physics and Technology, Moscow, Russia
[6] Sber AI Lab, Moscow, Russia

## ABSTRACT

Many tasks in graph machine learning, such as link prediction and node classification, are typically solved using representation learning. Each node or edge in the network is encoded via an embedding. Though there exists a lot of network embeddings for static graphs, the task becomes much more complicated when the dynamic (*i.e.,* temporal) network is analyzed. In this paper, we propose a novel approach for dynamic network representation learning based on Temporal Graph Network by using a highly custom message generating function by extracting Causal Anonymous Walks. We provide a benchmark pipeline for the evaluation of temporal network embeddings. This work provides the first comprehensive comparison framework for temporal network representation learning for graph machine learning problems involving node classification and link prediction in every available setting. The proposed model outperforms state-of-the-art baseline models. The work also justifies their difference based on evaluation in various transductive/inductive edge/node classification tasks. In addition, we show the applicability and superior performance of our model in the real-world downstream graph machine learning task provided by one of the top European banks, involving credit scoring based on transaction data.

# INTRODUCTION

It is crucial for banks to predict possible future interactions between companies: knowing that one company will be a client of another allows offering financial and other services. It is also important to have comprehensive and meaningful information about each client. If this knowledge is expressed as client embeddings, then the problem of their compactness and expressiveness emerges. Banks own large datasets of financial interactions between their clients, which can be used for training and testing models solving graph-related problems like prediction of future transactions or fraud detection.

Corresponding authors
Ilya Makarov, iamakarov@hse.ru
Andrey Savchenko, avsavchenko@hse.ru

Graph structures that describe dependencies between entities are widely used to improve the effectiveness of machine learning models trained on streaming data. In order to use conventional machine learning frameworks, it is necessary to develop a vector representation of the graph (such as network embeddings) by combining attributes from nodes (labels, text, etc.) and edges (*i.e.,* weights, labels, timing, interaction context) and taking into account the dynamic graph structure appearing in real-world problems.

In recent years, existing graph embedding models have been actively studied to apply deep learning methods for network representation learning. Hundreds of graph embeddings models have been developed and applied to different domains, especially in computer vision, text processing, recommendation systems, and interdisciplinary research in biology and genetics (*Makarov et al., 2021*). All approaches are united by a common problem statement, which is to learn an encoder model for the selected type of networks and important graph statistics. This will make it possible to apply standard machine learning frameworks and at the same time generalize attribute and structural information from the data.

Recently, modern machine learning methods have been proposed for processing networks and solving various prediction tasks on the local level (*Kipf & Welling, 2017*; *You et al., 2018*; *Veličković et al., 2017*; *Ding, Tang & Zhang, 2018*). Examples include node classification and link prediction for nodes both seen and unseen during model training. In practice, they represent important problems on large dynamic network data. Transaction data between companies and bank clients can help predict future transactions, user search history on the Web can be used to generate contextual advertising instances, disease transmission data can be used to predict epidemics dynamics.

Although most previous works on graph representation learning mainly focus on static graphs (with a fixed set of nodes and edges), there are many real-world applications in which graphs evolve over time, like social networks or sales data. One particularly common sub-type of graphs used to represent such structures is a temporal graph. It is a graph in which each edge has a time index, indicating a moment in time when the interaction, represented by an edge, occurred.

There are various problems when switching from static to dynamic networks (*Rosenberg, 1981*), among which computational complexity and variance of connectivity patterns over time (previous models could only exploit temporal difference statistics in the features domain like in works by *Makarov, Bulanov & Zhukov (2016)*; *Makarov et al. (2017)*; *Makarov et al. (2019)* or tackle temporal information in terms of missing link prediction (*Makarov et al., 2018*; *Makarov & Gerasimova, 2019b*; *Makarov & Gerasimova, 2019a*). In addition, for practical applications, one needs to have models suitable for inference in inductive settings which enable proper prediction on the fly with rear overall model retraining for large network data (*Kazemi et al., 2020*).

In this work, we describe a novel network embedding that combines the best elements of the efficient Temporal Graph Network embedding (TGN) (*Rossi et al., 2020*) and Causal Anonymous Walks (CAW) (*Wang et al., 2021b*). We choose the TGN as a backbone because it generalizes most existing temporal graph embedding models *via* flexible modular architecture. It allows updating node memory in a fast and expressive manner. Also, more

modern models like APAN (*Wang et al., 2021a*) or HiLi (*Wang et al., 2021a*) follow a similar paradigm of passing messages through the memory module. The CAW provides an opposite view on the graph evolution problem. It rejects the idea of memory and anonymizes each node. Instead, it aimed to build such a model, which can implicitly exploit laws of specific graph evolution ignoring node identities. As a result, CAW is unable to create node embedding but can significantly impact model quality by taking into account changes in graph structure. Fusion of two opposite ideas allows to build more precise network encoding methods.

To properly evaluate the proposed model, a unified experimental framework for temporal graphs is required. Most of the papers use different approaches to stream the graph events, mask nodes and edges and sample negative examples. We propose the pipelines for evaluation temporal embedding techniques in downstream graph machine learning tasks. It allows flexible integration of various models and different temporal network data under a unified evaluation framework shown in Fig. 1.

Our main contributions with this work consist of the following:

1. Novel temporal network embedding model achieving state-of-the-art results in various temporal graph machine learning tasks;
2. Standardized temporal network embedding evaluation framework and comparison of state-of-the-art models under common training setting, providing new insights and clarification of real-world performance compared to reported in the original research articles.

In addition, we prove the effectiveness of the proposed pipeline and its sub-modules *via* extensive ablation study and provide the industrial application of the proposed approach involving the transaction data of a major European bank. We showed that feature enrichment of temporal attention over temporal edge random walks improves quantitative and qualitative results in the real-world application of machine learning tasks on a banking graph.

## PRELIMINARIES & PROBLEM STATEMENT

In order to proceed with the problem statement, we describe basic concepts used throughout the text following notations by *Kazemi et al. (2020)*. We use $\mathcal{G}(\mathcal{V}, \mathcal{E})$ to denote a static graph, where $\mathcal{V}$ is the set of its vertices, and $\mathcal{E}$ is the set of edges of the graph, and $\mathcal{A}$ is an adjacency matrix of that graph.

A dynamic graph, in general, is a graph, which structure and node/edge attributes change over time. Generally, events may contain an updated state of the node. However, in our experiments, we consider all node features to be static and represent them as a matrix $\mathcal{X}$ with $|V|$ rows and $k$ columns, where $k$ is the node feature dimensionality.

In this work, we will use *dynamic graph* and *temporal graph* as interchangeable terms. We outline two possible kinds of dynamic graph representations below. There are two standard views on temporal network representation as a data structure:

- Discrete-time dynamic graph (DTDG) or snapshot-like graph is a sequence of snapshots from a dynamic graph, sampled at regularly-spaced times. Formally, we define DTDG

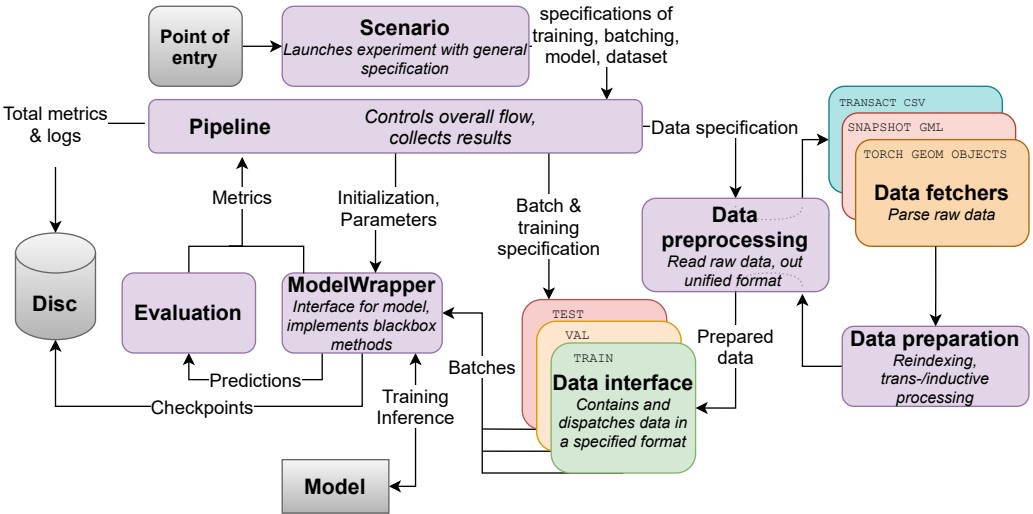

**Figure 1** Flowchart of the evaluation framework.

as a set $\{\mathcal{G}^1, \mathcal{G}^2, \ldots, \mathcal{G}^T, \}$, where $\mathcal{G}^t = \{\mathcal{V}^t, \mathcal{E}^t\}$ is the graph at moment $t$, $\mathcal{V}^t$ and $\mathcal{E}^t$ are sets of nodes and edges in $\mathcal{G}^t$, respectively.

- Continuous-time dynamic graph (CTDG) or transaction-/stream-like graph is denoted by pair $(\mathcal{G}, \mathcal{O})$, where G is a static graph representing an initial state of dynamic graph at time $t_0$, and $\mathcal{O}$ is event stream of events/transaction, denoted by tuple (*event_coordinates*, *event_data*, *timestamp*), where *event_coordinates* are ordered pairs of nodes, between which the event has occurred, and *event_data* is any additional data on the event.

Event stream may be represented as concatenation of index matrix containing transactions vectors $e_{ij}(t) = (v_i, v_j, t)$ with source and target node IDs, timestamp, and temporal edges' features.

We will refer to both CTDG and DTDG as dynamic graphs, although we will focus more on CTDGs as a natural representation of transactions in banking networks, which appear in non-uniform timestamps and represent a real-world streaming structured data, rather than discretized snapshot representation DTDG.

Finally, in our study, we consider such temporal graph machine learning problems as *node classification* and *link prediction*, both in transductive and inductive settings:

- *Transductive edge prediction* evaluates whether a transaction between two priorly known nodes occurred at a given time;
- *Inductive edge prediction* predicts a transaction between known and unknown nodes at a given time;
- *Transductive node classification* determines the dynamic label of a priorly known node;
- *Inductive node classification* determines the dynamic label of a priorly unknown node.

All the mentioned above problems on the chosen datasets can be formulated as binary classification with effortless extension to a multi-class case (*Savchenko, 2016*), which is evaluated *via AUC ROC* quality metric measuring performance for the classification at various error threshold settings.

# RELATED WORK

In this section, we overview state-of-the-art methods of constructing network embeddings for static and dynamic networks using taxonomies suggested by *Makarov et al. (2021)* and *Barros et al. (2021)*. We focus on dynamic network embedding as an evolutionary process of graph formation.

## Static graph embedding methods

When constructing network embeddings *via* solving the optimization problem, researchers usually focus on three main concepts: matrix factorization, node sequence methods, and methods based on deep learning. We consider the snapshot method, in which the current snapshot of a temporal network is taken, and missing links are predicted based on the available graph information.

Factorization techniques can be applied to different graph representations and optimized for different objectives, such as direct decomposition of the graph adjacency matrix (*Kruskal, 1978*; *Deerwester et al., 1990*; *Martinez & Kak, 2001*) or approximating proximity matrix (*Roweis & Saul, 2000*; *Tenenbaum, De Silva & Langford, 2000*). Despite factorizations being widely used in recommender systems, these models have high computational complexity and are difficult to extend for inductive learning.

Inspired by *Mikolov et al. (2013)*, sequence-based methods aim to preserve local node neighborhoods based on random walks. The two most prominent examples of models in this class are DeepWalk (*Perozzi, Al-Rfou & Skiena, 2014*) and Node2vec (*Grover & Leskovec, 2016*). Anonymous graphs walks have been proposed by *Ivanov & Burnaev (2018)*. However, their adaptations to temporal random walks have limited applications since they require retraining after adding new edges.

Recently, advances in geometric deep learning led to the creation of graph neural networks combining the best out of both fully connected and convolutional neural network architectures. Most of them use the method of neighborhood information aggregation from graph convolution network (GCN) (*Kipf & Welling, 2017*) and extend it with classical deep learning architectures such as recurrent neural network (RNN) (*You et al., 2018*), attention mechanism (*Veličković et al., 2017*), generative adversarial network (GAN) (*Ding, Tang & Zhang, 2018*), and graph transformers (*Yun et al., 2019*).

Recent studies show that a combination of deep learning models with semi-supervised techniques gives state-of-the-art results in terms of scalability, speed, and quality in downstream tasks (*Makarov et al., 2021*; *Makarov, Makarov & Kiselev, 2021*; *Makarov, Korovina & Kiselev, 2021*). However, static models are limited by the necessity to retrain the model with each significant change of graph structure.

## Dynamic graph embedding methods

Methods for dynamic graphs are often extensions of those for static ones, with an additional focus on the temporal dimension and update schemes (*Pareja et al., 2020*). All these techniques can be categorized according to which model of graph evolution representation is chosen: Continuous-Time Dynamic Graphs(CTDG) or Discrete-Time Dynamic Graphs (DTDG).

### DTDG-focused methods

Most of the early work on dynamic graph learning focuses exclusively on discrete-time dynamic graphs. Such models encode snapshots individually to create an array of embeddings or aggregate the snapshots in order to use a static method on them *Sharan & Neville (2008)*. All DTDG models can be divided into several categories, according to their approach for dealing with the temporal aspect of a dynamic graph.

### Single-snapshot models

Static models are used on the graph snapshots to make predictions for the next one (DNE (*Du et al., 2018*), TO-GAE (*Bonner et al., 2018*), DynGEM (*Goyal et al., 2018*)). Another way of implementing this methodology, called TI-GCN (Time Interval Graph Convolutional Networks) *via* residual architectures. It was proposed in SemiGraph (*Hisano, 2018*) and TI-GCN (*Xiang, Xiong & Zhu, 2020*). Besides single snapshots, these works use information from network formation (*Hisano, 2018*) represented by edge differences of several snapshots.

### Multi-snapshot models

The authors of TemporalNode2vec (*Haddad et al., 2020*) and Dyn-VGAE (*Mahdavi, Khoshraftar & An, 2019*) propose to learn structural information of each snapshot by separate models. *Haddad et al. (2020)* suggest to compute individual sets of random walks for each snapshot in Node2Vec fashion and learn final node embeddings jointly, while in *Mahdavi, Khoshraftar & An (2019)* autoencoders for each snapshot were trained in a consistent way to preserve similarity between consequent graph updates.

*RNN-based models.* In contrast to previous methods, models in this category aim to capture sequential temporal dependencies mostly by feeding output node embeddings or graph structure of each snapshot into RNN. Thus, GCN is combined with long short-term memory (LSTM) in CD-GCN (*Manessi, Rozza & Manzo, 2020*), GC-LSTM (*Chen et al., 2018*), GCRN (*Seo et al., 2016*) or gated recurrent units (GRU) in T-GCN (*Zhao et al., 2019*), DCRNN (*Li et al., 2018*). Following these ideas, authors of TNA (*Bonner et al., 2019*) and Res-RGNN (*Chen et al., 2019*) added residual connections to propagate topological information between neighboring snapshots. Recently, some papers (GCN-GAN (*Lei et al., 2019*), DynGAN (*Maheshwari et al., 2019*)) have proposed to use GANs in combination with RNNs. On the other hand, EvolveGCN (*Pareja et al., 2020*) argues that directly modeling dynamics of the node representation will hamper the model performance on graphs with dynamic node sets. Instead of treating node features as the input to RNN, it feeds the weights of the GCN into the RNN. *Hu et al. (2020)* proposed a model that balances between long- and short-term temporal patterns. It encodes long-term dynamics

by a GNN over a static graph and short-term dynamics by LSTM over several most recent node neighbors.

*Temporal graph attention.* Inspired by advances in natural language processing (NLP), models in this class leverage attention-mechanism to capture temporal information. Authors of A3T-GCN (*Bai et al., 2020*), LRGCN (*Li et al., 2019*), HTGN (*Menglin et al., 2021*) and DyHATR (*Xue et al., 2020*) follow the RNN module by an attention mechanism to take into account the contribution of its hidden states, while DySAT (*Sankar et al., 2020*) leverage self-attention mechanism without RNN stage.

*Convolutional models.* Although previous models can capture sequential dependencies, in practice, most of them use a limited number of historical snapshots. So, convolutions are used to propagate information between snapshots in STGCN(*Yu, Yin & Zhu, 2018*), TemporalGAT (*Fathy & Li, 2020*), MTNE (*Liu et al., 2021*).

Despite promising results, most of the models struggle from two disadvantages. First, methods lose the order of edge formation and reflect only partial information on network formation (*Xiang, Xiong & Zhu, 2020*). Second, computing static representations on each snapshot is inefficient in terms of memory usage on large graphs (*Cui et al., 2021*) and can not be used in practical applications.

### CTDG-focused methods

Continuous-time dynamic graphs require different approaches as it becomes computationally difficult to work with the entirety of such graphs after each interaction (*Goel et al., 2019*). Below we provide a more general classification of CTDG-focused methods, comparing with DTDG-focused ones based on approaches used for learning evolution patterns.

*Temporal random-walks models.* The approach implies including the time dependency directly in a sequence of nodes generated by random walks. Such methods create a corpus of walks over time (so-called "temporal random walks") with respect to the order of nodes/edges appearance in the graph. Based on this idea, authors of EHNA (*Huang et al., 2020b*) leverage a custom attention mechanism to learn the importance between a node and its temporal random-walk-based neighbors. CTDNE (*Nguyen et al., 2018*) proposes several methods to select the subsequent nodes connected to a starting one. A promising method for link prediction task was proposed by the authors of CAW (*Wang et al., 2021b*) who have developed Causal Anonymous Walks (CAWs), constructed from temporal random walks. CAWs adopt a novel anonymization strategy that replaces node identities with the hitting counts of the nodes based on a set of sampled walks to keep the method inductive. The model outperforms previous methods in both inductive and transductive settings. However, it is aimed to catch the correlation information between interacting nodes to learn temporal motifs. Such property is essential to catch the graph structure evolution specificity. Nevertheless, it does not construct the node embedding required by the practical downstream tasks like fraud detection.

*Local neighborhood models.* When interactions happen (node or edge adding and removal), models in this class update embeddings of the relevant nodes by aggregating information from their new neighborhoods. In DyGCN (*Cui et al., 2021*) and TDGNN

(*Qu et al., 2020*), the authors utilize a GCN-based aggregation scheme and propagate changes to higher-order neighbors of interacting nodes. To cope with information asymmetry, the authors of HiLi (*Chen et al., 2021*) propose to determine the priority of the nodes that receive the latest interaction information. JODIE (*Kumar, Zhang & Leskovec, 2019*) and TGAT (*Xu et al., 2020*) embed dynamic network for recommendation systems in similar ways. MNCI (*Liu & Liu, 2021*) uses additional temporal embedding of node's community aggregating it with node's neighborhood embedding *via* GRU. Several recent works(DyRep (*Trivedi et al., 2019*), LDG (*Knyazev, Augusta & Taylor, 2019*), $M^2$DNE (*Lu et al., 2019*)) consider interactions between nodes as stochastic processes with probabilities depending on the network statistics.

*Memory-based models.* The core idea of this class of temporal network embeddings lies in extending existing models by using special memory modules storing a history of interactions for each node. Methods vary from updating LSTM (DGNN by *Ma et al. (2020)*) to using augmented matrices of interactions (TigeCMN by *Zhang et al. (2020)*, designed for classification of nodes with labels fixed over time and thus, not suitable for our general-purpose framework). APAN (Asynchronous Propagation Attention Network) model (*Wang et al., 2021a*) aims to store detailed information about k-hop neighborhood interactions of each node in so-called ''mailboxes''. The recently developed Temporal Graph Network (TGN) proposed by *Rossi et al. (2020)* proposes a flexible framework, which consists of several independent modules and generalizes other recent CTDGs-focused models such as TGAT (*Xu et al., 2020*), JODIE (*Kumar, Zhang & Leskovec, 2019*), DyRep (*Trivedi et al., 2019*). Combining the advantages of JODIE and temporal graph attention(TGAT), TGN introduces the node-wise memory into the temporal aggregate phase of TGAT, showing state-of-the-art for industrial tasks results.

Because of the potential of CTDG-based models (*Gao & Ribeiro, 2021*) and necessity to apply the model to transaction data, we focus on developing a model in this class, while keeping in mind efficient DTDG models (*Xiang, Xiong & Zhu, 2020*). In what follows, we describe the idea of improving existing state-of-the-art CTDG models by properly fusing memory-based, neighborhood and interaction information under unified framework, thus combining multiple best practices of CTDG methods.

## PROPOSED APPROACH

As stated above in our literature survey, the best-known model for temporal graph prediction tasks, namely, TGN (*Rossi et al., 2020*), relies primarily on propagating information (''message'') through edges and generates node embeddings from occurring edges in a straightforward manner. At the same time, it was noticed that the CAW (*Wang et al., 2021b*) shows excellent performance in link prediction tasks by learning edge representations. Moreover, CAW encodes temporal network motifs, which can be possibly used to explain its predictions being an important property of the model as mentioned by *Holzinger et al. (2021)*. Unfortunately, CAW cannot be directly applied to extract node embeddings and, consecutively, downstream graph machine learning tasks, such as node classification.

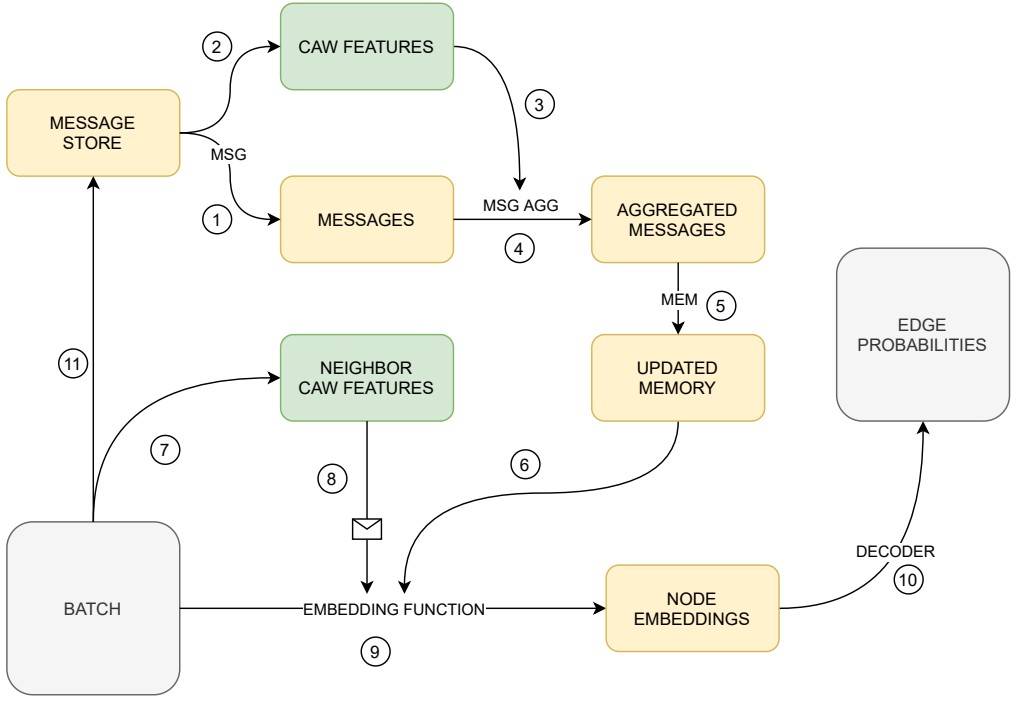

**Figure 2** **The proposed model.** Numbers denote the order of steps.

Hence, in this paper, we propose a novel network embedding taking the best out of the TGN framework and the CAW edge encodings (Fig. 2) to improve the quality of edge and node classification. Our contribution is to leverage the highly informative edge embedding generated in CAW. This will allow us to refine various functions of the TGN framework, including message extractor, embedding module, and memory aggregator.

## Model details

Below, we describe our model by listing and explaining all modules in a consecutive fashion. In the following descriptions and equations, LSTM refers to the Long Short-Term Memory type RNN, GRU is the Gated Recurrent Unit, "attn" and "self-attn" is attention and self-attention, respectively. The temporal random walk is crucial to handle dynamic networks. It is represented as a series of node-time pairs started from the specific node $i$ and sorted by time in descending order. The consequent pairs represent the temporal edge in a dynamic graph. Formally, it could be denoted by

$$W(i, M) = \{(w_0, t_0), (w_1, t_1), \ldots, (w_M, t_M)\}, w_0 = i, t_0 > t_1 > \cdots > t_M,$$

$$(w_{m-1}, w_m, t_m) \in \mathcal{E}, m \in \{1, 2, \ldots, M\}. \quad (1)$$

where $w_m \in \mathcal{V}$ is the node entered at step $m$, $t_m$ is the time of step $m$, $(w_m, t_m)$ is the $m$-th node-time pair. In future, we will omit the parameters $i, M$ for simplicity and refer to the elements of the $m$-th step as $W_m^{(1)} = w_m$ and $W_m^{(2)} = t_m$.

The temporal $k$-hop neighborhood of $i$th node before time $t$ is a set of nodes which can be reached from $i$ in $k$ steps *via* traversing in depth along the edges existing prior to time $t$, without regard for their direction. We will refer this set as $\eta_i^k$. The traversal starts from $w_0 = i$ visiting at most $k+1$ nodes in each walk. Thus, $w_j \in \eta_i^k$, if there exists a walk in the traversal from $w_0$ to $w_j$. More formally, $\eta_i^k$ can be defined as follows:

$$\eta_i^k = \{(w,t) \in W(i, k+1), \forall W\}. \tag{2}$$

### Neighborhood Edge Feature (NEF) generator

This trainable module and its integration in the following modules are intended to improve the predictive capabilities of the original TGN framework, which is our core contribution. It generates highly informative feature representations of pairs of nodes for any given moment in time. For a pair of nodes $i$ and $j$, the output of this module as $NEF_{ij}(t)$ is computed as follows.

At first, we sample an equal number $K > 1$ of the time-inverse walks for both nodes in order to capture information about the edge neighborhoods. All sampled walks have identical lengths (1–2 steps typically). A constant decay hyperparameter, which regulates how strongly the sampling process prioritizes more recent connections, is used to sample all walks.

The walk sets $S_i$ and $S_j$ are generated for edge $ij$ between nodes $i$ and $j$.$S_i$ consists of sampled time-inverse walks $W_k, k \in \{1, \ldots, K\}$. Each edge for each walk is sampled with the probabilities proportional to $exp(\alpha \Delta t)$, where $\alpha$ is the decay parameter, $\Delta t = t - t_p < 0$ ($t, t_p$ are the timestamps of the edge under sampling and the edge previously sampled, respectively).

The next step makes the sampled walks anonymous to limit extracted information to the edge neighborhood alone. Each node from $S_i$ is replaced by a pair of vectors that encode a positional frequency of the node in each corresponding position of walks in $S_i$ and $S_j$. The encoding vector of positional frequencies relative to the walks of $i$ for node $w$ is defined as:

$$g(w, S_i) = \{|\{W|W \in S_i, w = W_m^{(1)}, m \in \{1, \ldots, M\}\}|\}. \tag{3}$$

This equation simply specifies that the node index $w$ is encoded as a vector, so that each its $m$-th component is equal to the number of times where $w$ is the $m$-th node of some walk in $S_i$.

The anonymization of walks (*Wang et al., 2021b*) is achieved by using the defined function to transform node indices in walks. Each position in any walk of $S_i$ containing $w$ is replaced by

$$I^{ij}(w) = \{g(w, S_i), g(w, S_j)\}. \tag{4}$$

Similarly, any value in walks of $S_j$ filled by $w$ is replaced with $I^{ji}(w)$. In the remaining part of this section, we write $I = I^{ij}$, assuming the known orientation of the edge.

The remaining steps simply attempt to transform the "anonymous walk" representation of the node pair to a more compressed and usable state. Each obtained representation

of each position of each walk, *i.e.*, pairs of $\{g(w, S_j), g(w, S_i)\}$, are fed through separate instances of the same two-layered multi-layered perceptron (MLP) and sum-pooled:

$$f_1(I(w)) = \text{MLP}(g(w, S_i)) + \text{MLP}(g(w, S_j)). \tag{5}$$

Then, this representation is concatenated with the time-difference encoding and node or edge features of the corresponding step:

$$h(I(w)) = \text{concat}(f_1(I(w)), f_2(\Delta t), X) \tag{6}$$

where $f_2(\Delta t)$ is time Fourier features, and $X$ is a concatenation of all relevant node and edge attributes of the corresponding walk step. Finally, each walk with encoded positions is passed to an RNN (usually, Bi-LSTM):

$$\text{enc}(W) = \text{Bi-LSTM}(\{h(I(w_m)), m \in \{1, \dots, M\}\}), \tag{7}$$

where $w_m$ being $m$-th position of walk $W$.

The encoded walks are aggregated across all walks for the node pair $ij$:

$$NEF_{ij}(t) = \frac{1}{|S_i \cup S_j|} \sum_{W \in (S_i \cup S_j)} \text{agg}(\text{enc}(W)), \tag{8}$$

where $|S_i \cup S_j|$ is amount of walks in a set $S_i \cup S_j$, and agg is either self-attention module or identity for mean pooling aggregation.

### Message store

In order to apply the gradient descent, a memory of a node should be updated after it is passed as a training instance. Let's say an event associated with the $i$th node has occurred, *e.g.*, an edge involving $i$ has been passed for training in the current batch. Then, all information about the batch transactions involving node $i$ will be recorded in the message store after the batch inference, replacing the existing records for $i$. This information is used and updated during the processing of the next batch involving $i$.

### Message generator

When the model processes a batch containing a node, all transaction information about edges associated with this node is pulled. For each transaction between $i$ and some other node $j$ at a time $t$, a message for a node $i$ is computed as a concatenation of the current memory vectors of the nodes, edge features, time-related features, and neighborhood edge features of the corresponding edge:

$$m_i(t) = concat(s_i(t^-), s_j(t^-), t - t^-, e_{ij}(t), NEF_{ij}(t)), \tag{9}$$

where $ij$ is the index of a transaction between nodes $i$ and $j$, $s_i(t^-)$ is a memory state of node $i$ at time $t_i^-$ of last memory update for node $i$, and $NEF_{ij}(t)$ is a neighborhood edge feature vector for edge $ij$. The usage of NEF features here is our novel idea aimed to include more information about the type of the update message. The generated messages will update memory states of the batch nodes. Note that, aside from the basic and non-trainable concatenation, other choices for the message function, like MLPs are possible.

### Message aggregator

To perform a memory update on a node at time $t$, its message representation is obtained by aggregating all currently stored messages timestamps $t_1 < t_2 < \ldots < t$ which are related to this node:

$$\overline{m}_i(t) = \text{aggr}(m_i(t_1), m_i(t_2), \ldots, m_i(t)). \tag{10}$$

Here aggregation function *aggr* can be computed as a mean of generated messages. It is still possible to use the most recent message $m_i(t)$ as a value of $\overline{m}_i(t)$.

### Memory updater

The message generator and aggregator let the model encode useful transaction information as memory vectors. In particular, the memory state vector for node $i$ is updated at time $t$ by applying an RNN-type model to the concatenation of the received message and previous memory state:

$$s_i(t) = \text{RNN}(\overline{m}_i(t), s_i(t^-)) \tag{11}$$

where RNN is either GRU or LSTM, and initial state $s(0)$ is initialized with a random vector.

### Embedding generator

This module generates the node embedding based on memory states of the node and its k-hop neighborhood, features of the neighborhood, and NEF representations of "virtual" edges between the node and its direct neighbors. We include NEF features only to the direct neighbors due to its computational complexity. However, this does not affect our model because the first-order neighbors are sufficient. Resulting node embeddings can be viewed as autonomous node representations for classification (*Savchenko, 2017*). We propose, similarly to message generator, to add NEF features to node $i$ in order to let the model better discriminate neighbors on their relevance or type:

$$
\begin{aligned}
z_i(t) = &\sum_{j \in \eta_i^1} h_0(s_i(t), s_j(t), e_{ij}, v_i(t), v_j(t), NEF_{ij}(t)) \\
&+ \sum_{j \in \eta_i^k, k > 1} h_1(s_i(t), s_j(t), e_{ij}, v_i(t), v_j(t)),
\end{aligned} \tag{12}
$$

where $v_i(t)$ is a feature vector of node $i$ at time $t$, and $h_0$ and $h_1$ are the units of the neural network. It is typical to use MLPs as $h_0$ and $h_1$, but the following concatenation by self-attention is more preferable to capture complex dependencies involving NEFs.

### Embedding decoder

This is the final module, which transforms node embeddings into prediction results for downstream tasks. In this paper, we always use MLP with 3 layers with only node embeddings as inputs, and sigmoid or softmax output layers. For example, the multi-class classification problem (*Savchenko, 2016*) for node $i$ at time $t$ is solved using the following equation:

$$out_i(t) = \text{Softmax}(\text{MLP}(z_i(t))). \tag{13}$$

Similarly, the edge prediction task is solved as follows:

$$out_{ij}(t) = \text{Sigmoid}(\text{MLP}(\text{concat}(z_i(t)z_j(t)))). \qquad (14)$$

### Core idea and novelty of proposed model

Let us provide a high-level description of the model (Fig. 2). It learns to generate a temporal embedding for each node and decode embeddings into inputs for each classification task. The model assigns a memory vector to each node and generates each node embedding by aggregating memory vectors and other relevant features in a neighborhood of the node. Node memory vectors describe relevant information about interactions involving the node. They are updated using specified node messages, which, in turn, encode information about the last transaction involving the node. This design allows employing gradient descent for training memory and message generating modules. A decoder MLP transforms a pair of node embeddings into the probability of a temporal edge existing between the nodes. Similarly, it can be trained to transform a single node embedding into probabilities of the node belonging to each of the existing classes for a node classification problem.

There are two main modifications of the proposed model compared to the baseline TGN framework (*Rossi et al., 2020*). First, we change the message generating function, which provides the model with an additional way to differentiate the messages based on their relevance. The NEF features of an edge contain information about the walk correlation of the two nodes. As it can be used for very accurate link prediction (*Wang et al., 2021b*), it may be also used to classify some messages as being irrelevant, and diminish their effect on the memory update.

Second, while the original version of the embedding module in *Rossi et al. (2020)* allows treating different k-hop neighbors of the node differently if using attention for aggregation, it might be beneficial to provide the NEF features of connections between the node and its closest neighbors. As a result, we again take into account the walk correlation between the node and its neighbor, so that the differences in neighborhood type can be more evident.

## EVALUATION FRAMEWORK

In this section, we discuss our research methodology. It involves the evaluation framework, evaluation pipeline training settings, hyper-parameter choice and description of temporal networks used for the evaluation.

### Pipeline

The main contribution of the proposed framework (Fig. 1) is an easy-to-use unified data processing toolkit for an accurate evaluation of temporal network embeddings in downstream tasks under common training settings, which allows removing contradictions in experimental results reported in research articles in the field. The source code of this tool is publicly available (https://github.com/HSE-DynGraph-Research-team/DynGraphModelling). It is focused on transforming any graph into a universal format that is afterwards fed into the pipeline in either DTDG or CTDG format with the following features:

- selection of precise batching options;
- preparing data for both inductive and transductive settings (with support for bipartite data);
- interfacing with any kind of graph embedding models using an interface for model-framework communication, treating network embedding models as a black-box, which is then passed on to such basic methods as *train_model*, *predict_edge_probabilities*, etc.

## Training settings
### Negative sampling
The link prediction problem could be stated as a binary classification task. The general idea is to predict the presence of positive edges (which are present in the network) and the absence of negative ones (which represents non-connected pairs of nodes) equally well. However, most real-world networks are sparse. So the number of absent edges are significantly higher than the existent ones. It leads us to the imbalanced classification problem (*Savchenko, 2016*). The conventional way to handle it is to use the negative edge sampling strategies. It reduces the number of prevalent class examples to balance the representation of classes. We follow a standard setting (the same as used in *Rossi et al. (2020)*) in which we randomly select one negative sample for each positive one. That is done by a uniformly sampling set of nodes, which are then considered as destination points of the edges; the sources of the edges stay the same. In the training phase, the sampler considers only the training set of nodes; in the validation phase, both training and validation sets of nodes are used; in the test phase, the sampler considers nodes from the whole (unmasked) graph.

### Batch specification
The model is trained by passing edges (which includes negative samples) in batches sorted in chronological order.

### Training with temporal data masking
We use 80%–10%-10% train-val-test split with randomized training sets, supported *via* several data masking schemes. *Node masking* hides nodes (as well as all connected edges) from training data. The masked nodes for transductive tasks are also removed from the validation and test data, while for inductive tasks, masked nodes remain in validation and test data. *Edge masking* removes a fixed percentage of random edges from the whole dataset. We use two options for both masking techniques: the percentage of masked information is 10% and 70%, representing dense and sparse training settings, respectively. In particular, we report the results for three combinations, in which at least one mask is large, namely 10%–75% node-edge masking, 75%–10% and 75%–75%.

### Runs and validation
Each model/setting/dataset combination was run 10 times with random seeds and node/edge masking.

## Comparison with the state-of-the-art models

To evaluate our model (Fig. 2), we compare it with two kinds of models: the baseline TGN framework architectures (configured as described in *Rossi et al. (2020)*) and the more recent models that use edge features and consider interaction patterns. We chose DyRep (supporting long-term and short-term time scales of graph evolution and scaling to large graphs) (*Trivedi et al., 2019*), Jodie (working with bipartite graphs using node embeddings for predictive tasks with a future time lag) (*Kumar, Zhang & Leskovec, 2019*) and TGAT (using functional time encoding technique) (*Xu et al., 2020*) as representatives of the models of the first type. *Wang et al. (2021a)*; *Chen et al. (2021)*; *Zhang et al. (2020)* propose other ways of extracting and propagating edge features than our model and are selected as possibly more potent. It is important to emphasize that the CAW model cannot be compared with here because it was created only for graph reconstruction tasks and cannot produce node/edge embeddings for the downstream tasks.

## Datasets

In our study, we focus on well-known benchmark datasets for temporal networks. Their descriptive statistics are presented in Table 1, taking into account specifics of computing statistics for bipartite graphs of user-item interactions (*Latapy, Magnien & Del Vecchio, 2008*).

*Labeled datasets.* Both *Reddit* (*Hamilton, Ying & Leskovec, 2017*) and *Wikipedia* (*Foundation, 2010*) datasets are bipartite graphs representing interactions between users (source) and web resources (target), like posting to a subreddit or editing a wiki page. Text content is encoded as edge features to provide context. Both datasets have a binary target, indicating whether a user was banned or not.

*Non-labeled datasets.* The *UCI* (*Cortez & Silva, 2008*) dataset contains a communication history for the students' forum. The *Enron* (*Leskovec et al., 2008*) dataset is constructed over internal e-mail communication of Enron company employees. The *Ethereum* dataset (https://github.com/blockchain-etl/ethereum-etl) contains a directed graph of Ethereum blockchain transactions, which were collected from larger public BigQuery dataset (https://cloud.google.com/bigquery/public-data); we collected a dataset containing all transactions, which occurred during a 9-hour span. All of the non-labeled datasets are non-bipartite, with no additional features for edges or nodes.

## ABLATION STUDY

In this Section, an ablation study of the proposed architecture (Fig. 2) is provided to explore the impact of each submodule. Below we specify several additional modules added to interaction features processing pipeline for temporal network embedding *via* NEF generator,which outlined in the 'Neighborhood Edge Feature (NEF) Generator':

(Msg) NEF-Message concatenating NEF features and messages;
(Emb) NEF-Embed generates embeddings with NEF features;

**Table 1** **Descriptive statistics for the datasets.** Left to right: whether the graph is bipartite, number of unique nodes (representing users and items) and edges, labels, average degree, number of edge updates per node as source/target, and setting for splitting the temporal network into batches for snapshot DTDG models.

| | Bipartite | Nodes | Edges | Positive labels | Avg. degree | Avg. temporal edges | Batch setting |
|---|---|---|---|---|---|---|---|
| **Reddit** | Yes | 10000/984 | 672447 | 366 | 61.37 | 67.24/683.38 | Daily |
| **Wikipedia** | Yes | 8227/1000 | 157474 | 217 | 17.19 | 19.14/157.47 | Daily |
| **Enron** | No | 185 | 125236 | – | 29.06 | 675.78 | 1% |
| **UCI** | No | 1900 | 59836 | – | 13.04 | 106.41 | 1% |
| **Ethereum** | No | 231288 | 300000 | – | 2.34 | 2.59 | 1% |

**Table 2** Ablation study on Reddit dataset for inductive edge prediction.

| Enabled modules | AUC-ROC |
|---|---|
| (Msg)+(Emb)+(RNN) | $0.894 \pm 0.034$ |
| (Emb) | $0.893 \pm 0.035$ |
| (Msg) | $0.888 \pm 0.042$ |
| (Emb)+(Msg) | $0.878 \pm 0.051$ |
| (Msg)+(RNN) | $0.876 \pm 0.047$ |
| (Emb)+(RNN) | $0.870 \pm 0.055$ |
| TGN baseline | $0.865 \pm 0.065$ |

(RNN)  NEF-LSTM including Bi-LSTM into NEF generator, instead of mean pooling across walk elements.

Additionally, we consider alternating three important NEF-related hyper-parameters:

- number of generated random walk NEF-samples;
- random walk depth;
- positional dimension of random-walk-related embeddings.

Table 2 contains AUC-ROC (area under curve for receiver operating characteristic) standard deviation and mean averaged 10 times. The task measured was inductive edge prediction with default hyperparameters. Rather large and representative Reddit dataset is used in ablation study. Below we provide performance metrics for seven different combinations of modules mentioned above, except "(RNN)", which is considered a standalone TGN with NEF features processing regularization. Here the resulting model significantly improves the performance of standalone modifications and TGN baseline.

Table 3 depicts the average precision according to different choices parameters of CAW embedding module. All differences between scores are non-significant. So our model is stable between parameter choices. However, we recommend using neighbors from two hops with a small dimension of resulting node embedding to recover network structure more precisely and omit model over parametrization. Only a few samples of different random walks are enough.

**Table 3** The average precision of our model depends on parameters.

| Data | Number of walks | Hops | Dimension | AP |
|------|-----------------|------|-----------|-----|
| UCI | 32 | 1 | 10 | $0.759 \pm 0.000$ |
| UCI | 32 | 1 | 100 | $0.757 \pm 0.009$ |
| UCI | 8 | 2 | 10 | $0.764 \pm 0.001$ |
| UCI | 8 | 2 | 100 | $0.767 \pm 0.010$ |
| Wikipedia | 32 | 1 | 10 | $0.898 \pm 0.012$ |
| Wikipedia | 32 | 1 | 100 | $0.909 \pm 0.002$ |
| Wikipedia | 8 | 2 | 10 | $0.909 \pm 0.007$ |
| Wikipedia | 8 | 2 | 100 | $0.912 \pm 0.011$ |

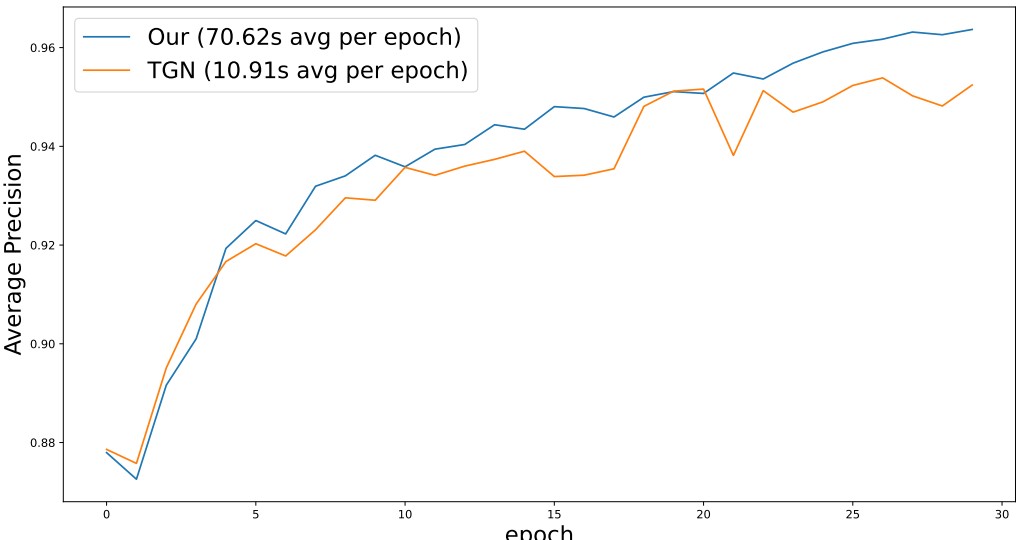

**Figure 3** Inductive average precision depends on the epoch number on Reddit dataset.

Figure 3 shows the dynamics of average precision for TGN and our models over a number of training epochs. Our model shows superior results after each epoch, requiring fewer data lookups to achieve competitive quality. However, due to the CAW extraction module, it works significantly slower (70.62s for our model and 10.91s for base TGN), which is consistent with *Wang et al. (2021b)* results. We leave speeding up CAW extraction module for future work.

## EXPERIMENTAL RESULTS

This section reports the mean AUC-ROC for our model and compares it with other state-of-the-art models described in 'Comparison with the state-of-the-art models'. Tables 4 and 5 show results for edge prediction task in transductive and inductive settings respectively. Best results are given in bold, while second best are underlined. "Node mask" and "Edge mask" columns specify the portion of nodes or edges used for model testing.

**Table 4  Transductive edge prediction, AUC-ROC.** Best results in bold, second-best underlined.

|  | Node mask | Edge mask | DyRep | Jodie | TGAT | TigeCMN | APAN | HiLi | TGN | Ours |
|---|---|---|---|---|---|---|---|---|---|---|
|  | 10% | 75% | 0.774 | 0.534 | 0.786 | – | 0.744 | 0.714 | 0.755 | **0.925** |
| Enron | 75% | 10% | 0.647 | 0.528 | 0.711 | – | 0.719 | 0.636 | 0.660 | **0.921** |
|  | 75% | 75% | 0.595 | 0.537 | 0.590 | – | 0.719 | 0.709 | 0.534 | **0.919** |
|  | 10% | 75% | 0.964 | 0.704 | 0.973 | 0.961 | 0.497 | 0.948 | 0.982 | **0.985** |
| Reddit | 75% | 10% | 0.973 | 0.825 | 0.962 | 0.824 | 0.502 | 0.961 | **0.970** | 0.968 |
|  | 75% | 75% | 0.959 | 0.720 | 0.936 | **0.962** | 0.500 | 0.950 | 0.956 | 0.920 |
|  | 10% | 75% | 0.770 | 0.496 | 0.835 | – | 0.858 | 0.546 | 0.878 | **0.982** |
| UCI | 75% | 10% | 0.731 | 0.573 | 0.826 | – | 0.867 | 0.556 | 0.872 | **0.977** |
|  | 75% | 75% | 0.781 | 0.489 | 0.834 | – | 0.855 | 0.539 | 0.776 | **0.960** |
|  | 10% | 75% | 0.966 | 0.728 | 0.970 | 0.824 | 0.556 | 0.863 | 0.978 | **0.983** |
| Wikipedia | 75% | 10% | 0.964 | 0.737 | 0.961 | 0.835 | 0.504 | 0.874 | 0.964 | **0.982** |
|  | 75% | 75% | 0.962 | 0.719 | 0.931 | 0.818 | 0.494 | 0.865 | 0.918 | **0.982** |
|  | 10% | 75% | 0.728 | 0.914 | 0.924 | – | **0.953** | – | 0.939 | 0.945 |
| Ethereum | 75% | 10% | 0.868 | 0.929 | 0.917 | – | **0.953** | – | 0.942 | 0.950 |
|  | 75% | 75% | 0.730 | 0.919 | 0.920 | – | **0.946** | – | 0.928 | 0.937 |

**Table 5  Inductive edge prediction, AUC-ROC.** Best results in bold, second-best underlined.

|  | Node mask | Edge mask | DyRep | Jodie | TGAT | TigeCMN | APAN | HiLi | TGN | Ours |
|---|---|---|---|---|---|---|---|---|---|---|
|  | 10% | 75% | 0.613 | 0.453 | 0.761 | – | 0.800 | 0.552 | 0.723 | **0.910** |
| Enron | 75% | 10% | 0.672 | 0.508 | 0.703 | – | 0.691 | 0.567 | 0.627 | **0.909** |
|  | 75% | 75% | 0.666 | 0.573 | 0.586 | – | 0.709 | 0.556 | 0.536 | **0.908** |
|  | 10% | 75% | 0.832 | 0.588 | 0.959 | 0.803 | 0.527 | 0.847 | 0.973 | **0.976** |
| Reddit | 75% | 10% | 0.910 | 0.294 | 0.960 | 0.644 | 0.511 | 0.928 | **0.967** | 0.966 |
|  | 75% | 75% | 0.880 | 0.298 | 0.933 | 0.680 | 0.494 | 0.924 | **0.955** | 0.915 |
|  | 10% | 75% | 0.602 | 0.576 | 0.833 | – | 0.675 | 0.546 | 0.825 | **0.969** |
| UCI | 75% | 10% | 0.723 | 0.431 | 0.816 | – | 0.745 | 0.517 | 0.821 | **0.967** |
|  | 75% | 75% | 0.718 | 0.423 | 0.829 | – | 0.750 | 0.601 | 0.775 | **0.954** |
|  | 10% | 75% | 0.950 | 0.584 | 0.964 | 0.644 | 0.500 | 0.824 | 0.969 | **0.977** |
| Wikipedia | 75% | 10% | 0.902 | 0.462 | 0.960 | 0.456 | 0.517 | 0.839 | 0.963 | **0.982** |
|  | 75% | 75% | 0.888 | 0.470 | 0.927 | 0.498 | 0.498 | 0.859 | 0.914 | **0.982** |
|  | 10% | 75% | 0.519 | 0.655 | 0.777 | – | 0.772 | – | 0.816 | **0.821** |
| Ethereum | 75% | 10% | 0.483 | 0.629 | 0.751 | – | 0.750 | – | 0.817 | **0.825** |
|  | 75% | 75% | 0.494 | 0.616 | 0.763 | – | 0.760 | – | 0.801 | **0.810** |

Table 4 present the results of edge prediction in the transductive setting. Our model outperforms others in all cases except for the Ethereum dataset. It is consistent with the reported results by *Wang et al. (2021a)* where the APAN model outperforms the TGN. On the given dataset, NEF features give only slight additional improvement compared to the vanilla TGN model. However, the APAN model seems to show poor performance when node and edge features are available. It requires matching the internal embedding dimension with external node and edge features. So it leads to the effective dimension mismatch that prevents APAN to converge in the best point. Also, our model is suboptimal

**Table 6 Node classification, AUC-ROC.** Best results in bold, second-best underlined.

| | Node mask | Edge mask | DyRep | Jodie | TGAT | APAN | HiLi | TGN | Ours |
|---|---|---|---|---|---|---|---|---|---|
| | 10% | 75% | 0.531 | 0.421 | 0.589 | 0.500 | 0.509 | 0.635 | **0.659** |
| Reddit (Transductive) | 75% | 10% | 0.601 | 0.435 | **0.665** | 0.500 | 0.491 | 0.584 | 0.627 |
| | 75% | 75% | 0.597 | 0.439 | 0.589 | 0.379 | 0.491 | 0.602 | **0.658** |
| | 10% | 75% | 0.510 | 0.456 | 0.555 | 0.500 | 0.551 | 0.576 | **0.625** |
| Reddit (Inductive) | 75% | 10% | 0.544 | 0.539 | 0.512 | 0.542 | 0.476 | 0.495 | **0.563** |
| | 75% | 75% | 0.521 | 0.567 | 0.435 | **0.603** | 0.524 | 0.592 | 0.584 |

on Reddit. It could be due to the dataset nature. Reddit is a user-post interaction, so the user's memory could be more important than NEF features, which aims to preserve how the whole graph evolves. The metrics for HiLi (*Chen et al., 2021*) on the Ethereum data are not presented in the table due to its inability to handle large graphs. It adds an identity matrix of node quantity dimension as an additional static vector, which leads to enormous memory consumption. The TigeCMN model is tested only on bipartite graphs because it requires different embeddings for users and items.

The results on the inductive edge prediction (Table 5) are consistent with the transductive one. The main difference of this setting is that the APAN model shows much worse results, and our model outperforms it on all of the datasets. Similarly to the previous scenario, TGN outperforms our model on Reddit.

Our model shows superior results on Reddit in transductive and inductive node classification settings in almost all cases (Table 6). The general problem could be that NEF features aim to capture the temporal motifs of the graph. So, when we pretrain our model on the link prediction problem, we left only a small variation for further adaptation to the node classification task.

In the end, despite some cases, our model consistently outperforms others. It seems that capturing network-wide common patterns with CAW-based features generally offers an improvement over existing models, although in some cases information added by these features may be duplicated by other encoding modules.

## BANKING RESULTS

As an industrial application of the proposed framework, we chose a pre-existing problem posited by a major European bank with a corresponding dataset that consists of transactions between different companies. In contrast to sequence-based methods for transactions analysis (*Babaev et al., 2019*; *Babaev et al., 2020*; *Fursov et al., 2021*), temporal network embedding provides new opportunities for analyzing relational activity between users and can benefit downstream machine learning tasks.

Each transaction has a timestamp so that we can treat it as a temporal edge. It allows us to consider only "edge events". We use the following edge features: transaction amount, timestamp and transaction type (*e.g.*, credit or state duty). We one-hot encode the type of transaction, which gives us 50 edge features in total: 49 from one-hot encoding and one from the transaction amount. These features were normalized.

At first, we compared our model with the original TGN (*Rossi et al., 2020*) on a link prediction task. We obtain the training and validation datasets consisting of transactions from a regional subdivision of a major European bank for about one week. This dataset contains about two million transactions. We select about 40,000 newest transactions to test the dataset, and all the rest is given to the train, so test dataset transactions are always newer than train transactions. We use the following procedure to check the performance of our model on the test dataset. First, we split training data into small chunks (400 batches) and gradually increase the number of chunks used to train the model. Each batch contains 512 transactions. The results are presented in Fig. 4.

We can see the very high performance of dynamic graph approach models. It can be noticed that the performance of our model grows a little bit slower than for TGN (*Rossi et al., 2020*), but after about 600,000 transactions, our model takes the lead. This slight time lag in training is due to our model taking additional time to learn extra information about interactions between nodes *via* learning CAW (*Wang et al., 2021b*) part of our model. This extra information allows gaining better results at the end of training.

The architecture of our model and TGN (*Rossi et al., 2020*) model allows us to produce node embeddings that can be used as input for a wide range of possible downstream tasks. For our experiment, we have taken the prediction of company default as the downstream task. The dataset contains about 5,000 companies, some of which will go bankrupt in a specific period in the future (180 days), and others will not. To train a classification model we used LightGBM (*Ke et al., 2017*). The obtained results were averaged over with 5-fold cross-validation and presented in Table 7.

We can see that our best model has 9.2% higher AUC ROC when compared to the TGN (*Rossi et al., 2020*). That means that extra information about interactions between nodes captured by CAW (*Wang et al., 2021b*) part of our model is useful for classification problems. We see also that the best performance is observed with a small CAW messages shape. This is related to the structure of our dataset. Its number of edges is comparable with the number of nodes. Thus the graph is sparse. Bigger CAW messages shapes can be helpful in more dense graphs.

## CONCLUSION

In this work, we proposed a novel model for temporal graph embedding (Fig. 2) that shows improvement over existing methods (*Rossi et al., 2020*; *Trivedi et al., 2019*; *Kumar, Zhang & Leskovec, 2019*) on various prediction tasks while preserving the ability to generate node embeddings. Moreover, we implemented a novel experimental framework (Fig. 1) that can process most kinds of graph data and an arbitrary dynamic graph inference model. The experimental study demonstrates the applicability of our method to solving various node/edge prediction tasks on temporal networks and significantly improving the existing results.

In future, it is possible to improve the performance of our framework for its application to the real-life temporal graph of bank's transactions with tens of billions of nodes. It is necessary to study efficient node/edge sampling strategies, choosing those that overcome

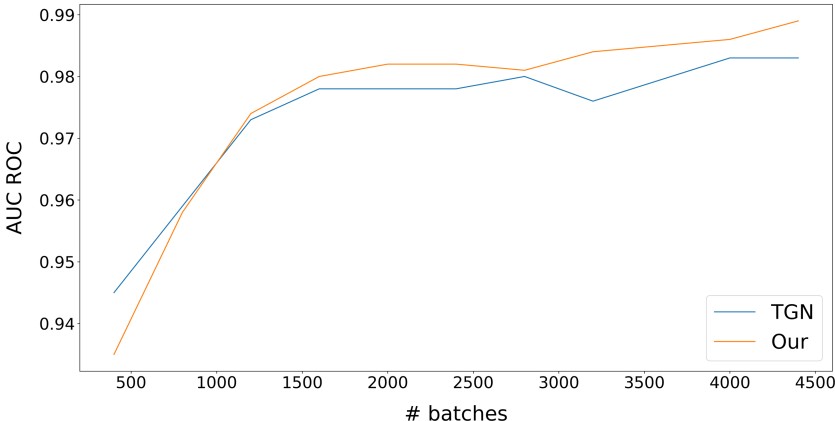

**Figure 4** Inductive edge prediction on transactions of regional subdivision of major European bank.

**Table 7** Node classification results based on node embeddings obtained from TGN and our proposed model with feature dimension $d$ being multiple of 8 × 2 shape.

| Node Embeddings | AUC-ROC |
| --- | --- |
| TGN | $0.621 \pm 0.066$ |
| Our | $0.678 \pm 0.059$ |

the limitations of current models when scaling to large graphs while preserving highly-informative edge features propagation.

## ACKNOWLEDGEMENTS

We thank all the colleagues and students who helps with the discussion of our study, in particular, Andrey Plyuschevskiy and Sergey Klyahandler.

### Funding

The work of Andrey Savchenko was supported by the Basic Research Program at the National Research University Higher School of Economics (HSE University). This research was also supported through computational resources of HPC facilities at HSE University. The funders had no role in study design, data collection and analysis, decision to publish, or preparation of the manuscript.

### Grant Disclosures

The following grant information was disclosed by the authors:
Basic Research Program at the National Research University Higher School of Economics (HSE University).
Computational resources of HPC facilities at HSE University.

## Competing Interests

Aleksandr Mikheev and Dmitrii Babaev are employees of Sber company.

## Author Contributions

- Ilya Makarov, Andrey Savchenko and Dmitrii Babaev conceived and designed the experiments, performed the experiments, analyzed the data, prepared figures and/or tables, authored or reviewed drafts of the paper, and approved the final draft.
- Arseny Korovko, Leonid Sherstyuk, Nikita Severin, Dmitrii Kiselev and Aleksandr Mikheev conceived and designed the experiments, performed the experiments, analyzed the data, performed the computation work, prepared figures and/or tables, authored or reviewed drafts of the paper, and approved the final draft.

## Data Availability

The benchmark data is available in the references and at GitHub: https://github.com/HSE-DynGraph-Research-team/DynGraphModelling/tree/main/data.

The code with data loaders is available at GitHub: https://github.com/HSE-DynGraph-Research-team/DynGraphModelling.

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
