# Peer review of "Temporal network embedding framework with causal anonymous walks representations"

_PeerJ Computer Science, doi:10.7717/peerj-cs.858_

## Round 0.1 · original submission · Major Revisions

A revision is needed before your manuscript can proceed. Please provide a detailed response letter. Please note that I do not expect you to cite any recommended references unless they are essential and relevant. I look forward to receiving your revision.

Reviewer 1 ·

Basic reporting

No comments. Language is in general clear. Sufficient background is given. Terms are defined before use.

Experimental design

Experimental designs make sense and cover a wide range of problems. Experimental details are sufficient.

Validity of the findings

The findings are valid. Experimental results support the findings.

Additional comments

N/A

Reviewer 2 ·

Basic reporting

Easy to understand, interesting, relevant and convicing.

Experimental design

well defined, relevant, meaningful and convincing.

Validity of the findings

The findings are well presented, easy to understand and convincing. Also the conclusions are well stated and clearly support the assumption given before.

Additional comments

As the authors correctly present, many machine learning tasks for graphs are increasingly prevalent, especially in link prediction and node classification, and indeed the challenge becomes greater when analyzing the dynamic (i.e., temporal) network. Therefore, in this paper, the authors propose a new approach to learn dynamic representations based on temporal graph networks by introducing a message generation function by extracting causal anonymous paths. To evaluate their novel approach, the authors present a benchmark pipeline for evaluating temporal network embeddings. Truly, this is a very interesting, relevant and also important approach. The authors also nicely demonstrate the applicability of their model in the real downstream graph of one of the leading European banks performing credit scoring based on transaction data - which is a good application.

This reviewer is very positive and recommends the acceptance of this article and only recommends to make a few minor improvements:

1) page 4, section 3.1. line 157, indeed graph neural networks are enormously important for this approach and some very good background work has also been listed, here for completeness a very recent and already highly cited related work should be cited and that is by Holzinger et al (2021), Towards multi-modal causability with Graph Neural Networks enabling information fusion for explainable AI, https://doi.org/10.1016/j.inffus.2021 .01.008 - this reference is in fact an important indication for the interested reader of the possibility of traceability and understandability (which is currently subsumed under explainability), which is also interesting for further work in the context of this application domain (baking, finance).

2) Figure 1 is very good, but the small text in the boxes (this reviewer printed the work on paper) is practically impossible to read. (Figure 2 is perfect)

3) Table 1 is also difficult to read

4) Figure 3 needs a legible caption and for faster reader acquisition a better caption. (Figure 4 again is good)

Generally a very good paper and this reviewer congratulates the authors.

Reviewer 3 ·

Basic reporting

The paper is well organized but the presentation could be improved. References can provide sufficient field background. The article structure, figures and tables are clear, and the raw data and code are also provided.
In addition, there are some problems with this paper.

Q1: The second half of Equation 2 is somewhat difficult to understand, and I hope to see a more intuitive definition of “the t-timed k-hop neighborhood”.

Q2: The calculation of time difference t (Line 274, Page 7/18) is not given, which can be further explained. If t is calculated directly, perhaps it means that the earlier the node appears, the higher the probability of being selected?

Q3: In Equation 12, there are two feature vectors, how are they obtained? In addition, what is the difference between the feature vectors at different times?

Q4: In Section 3, the paper gives abbreviations for some methods (such as DeepWalk, EvolveGCN and CAW), but not for others, and I suggest consistency in formatting.

Q5: There are also some new works can be introduced in Section 3, including but not limited to the following works.
[1] Jiancan Wu, Xiang Wang, Fuli Feng, Xiangnan He, Liang Chen, Jianxun Lian, and Xing Xie. 2021. Self-supervised Graph Learning for Recommendation. In Proceedings of the 44th International ACM SIGIR Conference on Research and Development in Information Retrieval (SIGIR '21).
[2] Meng Liu and Yong Liu. 2021. Inductive Representation Learning in Temporal Networks via Mining Neighborhood and Community Influences. In Proceedings of the 44th International ACM SIGIR Conference on Research and Development in Information Retrieval (SIGIR '21).
[3] Menglin Yang, Min Zhou, Marcus Kalander, Zengfeng Huang, and Irwin King. 2021. Discrete-time Temporal Network Embedding via Implicit Hierarchical Learning in Hyperbolic Space. In Proceedings of the 27th ACM SIGKDD Conference on Knowledge Discovery & Data Mining (KDD '21).
[4] Hu, Linmei and Li, Chen and Shi, Chuan and Yang, Cheng and Shao, Chao. 2020. Graph neural news recommendation with long-term and short-term interest modeling. Information Processing & Management.

Q6: A method abbreviation (such as DeepWalk) may help your paper spread more effectively.

Experimental design

The paper's research matches the aims and scope of the journal. Research question well defined, relevant, and meaningful. The paper gives the raw data and code, and provides enough details to reproduce the model. The paper conducts experiment on several datasets and discusses the experimental results to demonstrate the validity of the work.
In addition, there are some problems with this paper.

Q7: The paper conducts experiment on link prediction and node classification tasks in transductive and inductive settings. I would like to see the construction details for the downstream tasks described further.

Q8: In Line 348, Page 10/18, the paper mentions that “the number of negative samples is equal to the number of edges”. I am interested to see a further explanation of it.

Validity of the findings

The paper presents an innovative combination of CAW and TGN methods, and the experimental results demonstrate its effectiveness. All underlying data have been provided, and conclusions are well stated.
In addition, there are some problems with this paper.

Q9: The paper seems to embed CAW as a module in the structure of TGN, does it lead to a high complexity? I would like to see a discussion on the complexity of the model.

---

## Round 0.2 · accepted · Accept

This paper can be accepted. Congratulations!

Reviewer 2 ·

Basic reporting

The authors have taken into account the reviewer's comments and addressed them appropriately, and the reviewer would now recommend that the paper be accepted.

Experimental design

n.a.

Validity of the findings

n.a.

Additional comments

Of course run the usual final spell checks when going into production.

Reviewer 3 ·

Basic reporting

All my questions have been explained in the revision. No other comments.

Experimental design

Experimental designs make sense and cover a wide range of problems. Experimental details are
sufficient.

Validity of the findings

The findings are valid. Experimental results support the findings.

Additional comments

N/A